# Improvement of Atopic Dermatitis by Synbiotic Baths

**DOI:** 10.3390/microorganisms9030527

**Published:** 2021-03-04

**Authors:** Matthias Noll, Michael Jäger, Leonie Lux, Christian Buettner, Michaela Axt-Gadermann

**Affiliations:** Institute for Bioanalysis, Department of Applied Sciences, Coburg University of Applied Sciences and Arts, 96450 Coburg, Germany; michael.jaeger93@gmx.de (M.J.); leonie.lux@behrens-pm.de (L.L.); Christian.Buettner@hs-coburg.de (C.B.); michaela.axt-gadermann@hs-coburg.de (M.A.-G.)

**Keywords:** amplicon sequencing, atopic dermatitis, bacterial 16 S rRNA gene, co-occurrence networks, human skin microbiome, severity scoring of atopic dermatitis, staphylococcus

## Abstract

Atopic dermatitis (AD) is a widespread chronic inflammatory dermatologic disorder. This randomized, double-blind study aims to evaluate the effect of synbiotic baths with a defined mixture of six viable lactic acid bacteria (LAB) and prebiotics, without bacteria and prebiotics and placebo baths without prebiotics and bacteria to treat AD patients over a period of 14 days. Therefore, AD patients were randomly assigned into three groups using synbiotic (*n* = 7), prebiotics (*n* = 8) or placebo baths (*n* = 7). Severity of AD was evaluated over time by using severity scoring of atopic dermatitis (SCORAD) and by patient questionnaires. In addition, microbiome on eczematous skin surface was sampled by swaps from each patient before the bath treatment, and after 9, 11 and 14 days of bath treatment. Thereafter, nucleic acids were extracted and the bacterial 16S rRNA gene was amplified via PCR for subsequent amplicon sequencing. Results showed a significantly reduced SCORAD over time of AD patients after daily synbiotic or prebiotic baths. Moreover, AD patients after daily synbiotic baths had a significantly improved pruritus and skin dryness and their bacterial microbiome was enriched by LAB. Taken together, a synbiotic bath is a promising topical skin application to alleviate AD.

## 1. Introduction

Atopic dermatitis (AD) is one of the most common dermatologic disorders affecting up to 20% of children as well as 1 to 3% of adults worldwide [1]. Pathogenesis is impacted by a multitude of genetic and environmental factors such as skin barrier dysfunction [2], abnormal protein and enzyme processing [3,4], impairment of tight junctions [5], impairment of the inflammatory cascade [6], dysbiosis of the skin microbiome [7] and exposure to irritant or proven allergic substances [8]. Due to the multifactorial nature of AD, efficient and well-tolerated treatment without any severe side effects are still challenging.

Treatment of AD can be categorized into basic skin care, topical corticosteroids (TCSs) and calcineurin inhibitors (TCIs), phototherapy or balneo phototherapy and systemic therapies based on immunosuppressive therapies or biologic drugs. Basic skin care is characterized by the application of emollients in order to moisturize the skin, restore the natural epidermal barrier and prevent further water loss [9,10]. However, its efficacy in moderate and severe cases of AD is limited. TCSs are placed on the skin by ointments, creams, sprays and foams [11]. TCIs can be used to inhibit the pro inflammatory cytokine gene transcription [12]. Despite their obvious merits, TCSs and TCIs are often scrutinized due to adverse side effects, which include ocular diseases [13], corticophobia [14], cutaneous atrophy, telangiectasia, striae, steroid rosacea and perioral dermatitis, hypothalamic pituitary-adrenal axis suppression and skin infections [15]. Finally, phototherapy is based on narrowband ultraviolet light two to three times per week over a timeframe of up to 12 weeks. Due to limited efficacy, it is most often used as a supplemental treatment [16,17]. Current treatments focus on affected local areas, while systemic treatments have the ability to affect the entire body. Systematic treatment methodologies can cause untargeted suppression or modulation of components of the immune system [18].

Recent research indicate a correlation between AD severity and species distribution inside the microbiome of affected skin areas [19]. *Staphylococcus aureus* superinfection in particular was consistently linked with AD severity [20,21]. Based on these findings, manipulation of the skin microbiome via pre- and probiotics was considered as a potential treatment option for patients suffering from AD. Prebiotics are non-digestible ingredients that beneficially affect the host by stimulating growth of selected range of microorganisms [22]. Probiotics are defined as viable microorganisms that confer a health benefit on the host, when they are applied in adequate amounts [23]. Combinations of prebiotics and probiotics are known as synbiotics. Some studies indicate a reduction of AD occurrence and severity after oral probiotic treatments in infants and adults by a systematic treatment [24,25]. However, other studies showed no significant effect on AD after oral consumption of probiotics [26,27] indicating that studies including a great variety of patient anamneses to retrieve a generalized conclusions are still lacking. In turn, information of local application of synbiotics or prebiotics to treat skin of AD patients is still rare. Synbiotic baths as local AD treatment showed a significant improvement of the AD patients’ quality of life (QoL) [28]. These findings however rely on scoring of atopic dermatitis (SCORAD) and QoL measurements only. While these techniques are valuable tools, which allow profound statements of AD severity, no conclusions of changes in the skin microbiome can be stated.

This double-blind study aims to investigate the changes in the skin microbiomes of AD patients after local treatment with synbiotic, prebiotic or placebo baths as well as their correlation with patients’ SCORAD and QoL over time. To this end, these data were correlated with the bacterial microbiome, which was achieved by a 16S rRNA gene amplicon sequencing approach. Additionally, a co-occurrence network analysis of the amplicon sequencing data was performed according to Olesen et al. [29] in order to explorer bacterial co-presence and mutual exclusion patterns between specific bacterial operational taxonomic units (OTUs) and bath treatments.

## 2. Materials and Methods

### 2.1. Study Design

A double-blind study with 22 AD patients with atopic eczema on hands, arms, feet or legs was conducted. AD patients with an age less than five, pregnancy, immunodeficiency and/or ongoing antibiotic treatment were excluded prior to the start of the study. AD patients stopped other topical or systematic treatment seven days prior to the start of this study. AD severity of each patient was assessed by a dermatologist, and thereafter patients with similar AD severity were randomly assigned to three different treatment groups, which take a daily bath with a synbiotic (*n* = 7), prebiotic (*n* = 8) or placebo bath (*n* = 7). The synbiotic bath consisted of the probiotic strains 1 × 10^9^ Colony forming units (CFUs) L^−1^
*Bifidobacterium breve* (ATCC 15698), 1 × 10^9^ CFUs L^−1^
*Bifidobacterium animalis* subsp. *lactis* (ATCC 27536), 1 × 10^9^ CFUs L^−1^
*Lactobacillus casei* (ATCC 393), 1 × 10^9^ CFUs L^−1^
*Lactobacillus gasseri* (ATCC 33323), 1 × 10^9^ CFUs L^−1^
*Lactobacillus plantarum* (ATCC 14917) and 1 × 10^9^ CFUs L^−1^
*Lactobacillus rhamnosus* (ATCC 53103), which were purchased from the American type culture collection and used after culturing as lyophilized powder in the synbiotic bath. As 0.5 × 10^9^ CFUs of each strain was available in one g (six strains resulted six g including prebiotics), twelve gram per L were added and bath volume was approx. 4 L for legs and 3 L for arms. In addition, the synbiotic bath included 2.88 g L^−1^ maltodextrin, 6 g L^−1^ inulin, and 3 g L^−1^ apple pectin as prebiotics. The prebiotic bath contained only the prebiotics in the same concentration. Finally autoclaved sand was used as only ingredient for the placebo bath. Ingredients for each bath were separately added to hand-warm water and the skin surfaces with atopic eczema were bathed for ten minutes each day over a period of 14 days. After the bath remaining bath ingredients were dried on skin surface. The skin surface was documented by SCORAD as explained earlier [30], and sampled by cotton swaps (VWR International GmbH, Darmstadt, Germany) at the beginning of the experiment, and after nine, eleven and fourteen days of treatment. In addition, each AD patient assessed the own QoL by a questionnaire at the beginning of the experiment, after nine and fourteen days of treatment, which addressed the overall assessment, redness, pain, restriction, pruritus, lichenification and dryness of AD. Attending AD patients were asked to rate each parameter based on a scale ranging from one to ten with ten being the most severe by a questionnaire. To document any visible changes of the atopic eczema during the treatment period, photographs were taken at the beginning of the experiment, and after nine and fourteen days of treatment.

### 2.2. Nucleic Acid Extraction

Nucleic acids were extracted from the cotton swaps as described earlier [31] with minor modifications. Briefly, the head of the cotton swabs were cut with a sterile scissors and mixed with 0.2 g zirconium beads (Carl Roth GmbH, Karlsruhe, Germany), 400 μL cold TPM buffer (50 mM Tris HCl pH 6.8; 1.7% (*w*/*v*) polyvinylpyrollidone K25; 20 mM MgCl_2_), 200 μL NaPO_4_ buffer (200mM NaPO_4_ with pH 5.6) and 600 μL SDS-phenol (all chemicals from Carl Roth). The mixture was vortexed and incubated for 10 min at 65 °C. Thereafter, each sample was homogenized with a ball mill (FastPrep^®^-24, Thermo Fisher Scientific, Waltham, MA, USA) for 60s at 4.5 m s^−1^ and frozen for five minutes at −80 °C. Each sample was subsequently centrifuged for 15 min at 21.500× *g* and 4 °C. Supernatant (approx. 800 μL) was mixed with the same volume of TPM buffer and centrifuged again (21,500× *g*, 4° C, 15 min). Thereafter, 800 μL of supernatant was mixed with the same volume of phenol-chloroform-isoamylalcohol (PCI, Carl Roth) and centrifuged once more (21,500× *g*, 4° C, 15 min). 650 μL supernatant was mixed with 1300 μL PEG buffer (30% polyethylene glycol 6000 in 1.6 M NaCl-solution) and 2 μL glycogen (VWR International GmbH). The mixture was centrifuged for 45 min at 4 °C and 21,500× *g*. The supernantant was discarded. The resulting nucleic acid pellet was washed two times with ice-cold 500 μL 70% ethanol and dried at 37 °C. The pellet was then resuspended in 50 μL 1 × TE buffer (Carl Roth). Quality and quantity of nucleic acids was measured with Thermo Scientific™ Multiskan™ GO Microplate Spectrophotometer (Thermo Fisher Scientific) as outlined by the manufacturer.

### 2.3. PCR and Amplicon Sequencing

16S rRNA gene was amplified from each of the 80 nucleic acid extracts using the 341F (5′CCTACGGGNGGCWGCAG’3) and 785R (5′GACTACHVGGGTATCTAATCC’3) primer pair [32] as described earlier [33]. PCR amplicons were thereafter ligated to inline barcode sequences and subsequently 300 bp paired-end sequenced by Illumina Miseq V3 System (San Diego, CA, USA), which was carried out by LGC Genomics (LGC Genomics GmbH, Berlin, Germany). Sequence raw data were demultiplexed by using the Illumina bcl2fastq 2.17.1.14 software (Illumina, San Diego, CA, USA) and were subsequently sorted by reads of amplicon inline barcodes. Barcode sequences, adapters and primers were clipped from the sequence and forward and reverse reads were combined by using BBMerge 34.48 [34]. 16S rRNA gene sequences were pre-processed and operational taxonomic units (OTUs) were picked from the amplicons with Mothur 1.35.1 [35]. Sequences with ambiguous bases, with homopolymer stretches, short reads, chimera and with an average quality score below 33 were removed [35]. 16S rRNA gene sequences were aligned with the 16S Mothur-Silva SEED r119 reference database [35], and OTUs were picked by clustering at the 97% identity level and taxonomical classified against the Silva reference classification [36]. Two samples were discarded due to low sequencing read counts. Thereafter, singleton and doubleton OTUs were removed and a table with individual 937 OTUs was generated (Appendix A). The shannon diversity index, the simpson‘s diversity index and the Chao1 richness estimator were calculated for all 78 samples using the diversity and estimateR functions R-package vegan [37]. The data was visualized using the Origin 2019b program (Origin Lab Corporation, Northampton, MA, USA).

### 2.4. Statistics and Network Analyses

Normal distribution of the data set was tested for the 10 most abundant OTUs over time by the Shapiro–Wilk test [38] using the software Origin 2019b (OriginLab Corporation, Northampton, MA, USA). Two-way ANOVA was used to test the effect of SCORAD, QoL parameters, bacterial diversity indices over time and over bath treatment and over the interaction of time and bath treatment. The significance threshold was defined as *p* ≤ 0.05. Correspondence analysis was carried out with relative sequence read abundances as reported earlier [33].

Co-occurrence network analyses for each bath treatment were calculated by using cytoscape with the plugin CoNet [39,40]. OTU table of each treatment with absolute sequence counts were used, if OTUs were present in at least three AD patients of a treatment bath group. Network calculations were carried out with following methods: Pearson, Spearman correlation, Steinhaus similarity, Bray-Curtis and Kullback-Leibler dissimilarity. Minimal occurrence of observations for each set of replicates was set to at least 60% (i.e., 24, 15 and 9 observations for the different networks). Threshold was set to 3000 top and bottom edges, so that each correlation and dissimilarity method contributed 3,000 positive and 3000 negative edges to the initial network. MinSupport was selected to be three, so only edges supported by at least three of the five methods were kept. The method-specific *p*-values were computed by using the mean and standard deviation of the bootstrap distribution (100 iterations) as a parameter of the normal distribution. Method-specific *p*-values were then merged using the method of Brown [41]. Only edges with *p* < 0.05 were kept after multiple-testing correction of Benjamini and Hochberg [42]. In addition, the difference between all co-occurrence networks from synbiotic, prebiotic and placebo bath treatments was also calculated with the same plugin CoNet using the merge difference function. The resulting merged co-occurrence network enabled the examination of changes in network compositions between bath treatments. Nodes of the final networks were assigned to modules by using GLay community algorithm [43].

Finally, within-module connectivity (z) and among-module connectivity (Pi) were calculated as described by Guimera and Amaral [44] with an automated in-house excel sheet. Peripheral nodes (specialists) were defined by z ≤ 2.5 and Pi ≤ 0.62, connectors by z ≤ 2.5 and Pi > 0.62, module hubs by z > 2.5 and Pi ≤ 0.62 and network hubs by z > 2.5 and Pi > 0.62 as introduced earlier [29].

## 3. Results

### 3.1. Synbiotic Baths Improved SCORAD and QoL

The SCORAD of AD patients after 14 days of a daily synbiotic or prebiotic baths decreased significantly (Figure 1). Moreover, SCORAD of AD patients of synbiotic bath improved significantly better compared to the other bath treatments over time. In contrast, the SCORAD of AD patients after a daily placebo bath did not change over time.

In addition, the AD patients evaluated their skin appearance and their QoL. After all synbiotic baths, AD patients recognized significantly reduced pruritus and skin dryness over time (Table 1). In addition, overall assessment, restriction, redness, pain and lichenification improved over time for AD patients after synbiotic or prebiotic baths but not for AD patients after placebo baths. Moreover, photographs of the AD sides of patients after synbiotic or prebiotic baths showed improvements of the skin appearance (Appendix A).

### 3.2. Composition of the Skin Microbiome Was Shifted

The composition of the bacterial microbiome shifted over time and was more similar to each other from AD patients after synbiotic bath treatment compared to prebiotic or placebo bath treatment as these both treatments revealed a higher compositional variance (Figure 2). Members of the genus *Staphylococcus* were the most dominant taxa of the skin microbiomes of AD patients, which covered approx. one-third to two-thirds of the relative sequence read abundance of all samples (Appendix A). AD patients treated with synbiotic baths exhibited the highest relative sequence read abundance of members of the genus *Staphylococcus* before the bath treatment, which is in accordance to the severity reflected by the SCORAD (Figure 1). Although the SCORAD significantly decreased, the sequence read abundance of members of the genus *Staphylococcus* did not significantly alter over time (Appendix A).

Members of the genera *Enhydrobacter*, *Corynebacterium*, *Acinetobacter*, *Micrococcus*, *Escherichia* and *Paracoccus* were also abundant (Appendix A). Relative sequence read abundances of members of the genera *Lactobacillus* and *Bifidobacterium* increased from none to 8% and 6%, respectively, over time only by AD patients treated with synbiotic baths, which contained these species (Appendix A). Therefore, the viable microorganisms as part of the synbiotic bath ingredients became part of the bacterial skin microbiome already after nine days of bath treatment, which also affected the bacterial microbiome composition (Figure 2). AD patients treated with prebiotic baths were characterized by relative sequence read abundances of members of the genera *Micrococcus* and *Paracoccus* (Appendix A). In turn, AD patients treated with placebo baths showed an increase in relative sequence abundances of members of the genera *Staphylococcus*, *Paracoccus* and *Enhydrobacter* and a decrease of *Corynebacterium* over time (Appendix A). The diversity indices Shannon and Simpson’s and Chao1 richness estimator did not alter significantly over time in any bath treatment (Appendix A).

### 3.3. Bacterial Co-Occurrence Networks of AD Patients Differed

Three independent co-occurrence network analyses were individually calculated for AD patients after daily synbiotic, prebiotic or placebo bath treatments, which consisted of 2274, 1990 and 1960 significant interactions (edges) between 241, 338 and 338 different OTUs (nodes), and six, 13 and ten distinct modules, respectively. Members of the genus *Micrococcus* were identified as a network hub of AD patients after synbiotic bath treatment (Figure 3A). Network of AD patients after a prebiotic bath showed a non-cultured genus of the order Obscuribacterales as a module hub, and members of the genera *Caulobacter*, *Sphingopyxis*, *Aquabacterium* and *Chthoniobacter* as module hubs (Figure 3B, Table 2). Network of AD patients after a placebo bath showed the highest connectivity among the three networks, with four module hubs and 15 connectors (Figure 3C, Table 2). Notably, no significant interactions with the dominant members of the genus *Staphylococcus* were identified in any network analyses (Appendix A).

In addition, a merged network based on differences between these three co-occurrence networks was generated, which consisted of 1817 interactions of 337 OTUs between each bath treatment group (Appendix A). Therefore, substantial differences in interactions patterns between groups were illustrated. Members of the genus *Acidibacter* exhibited more interactions in the microbiomes of AD patients with placebo baths compared to AD patients that used synbiotic or prebiotic baths (Appendix A). Members of the genus *Propionibacterium*, had a more prominent topological role for AD patients after a prebiotic bath.

## 4. Discussion

The majority of AD patients had a long history of topical and systematic treatments to treat their AD. As AD patients has to stop their previous treatment seven days prior this study a typical rebound phenomenon was observed, which has been described before [45]. This rebound phenomenon affected the early phase of each bath treatment as the SCORAD after three and seven days was not significantly different to the initial SCORAD. Therefore, we chose 9, 11 and 14 days of each bath treatment for in depth microbiome analyses.

AD severity is very often linked to the abundance of *S. aureus* in the AD skin microbiome [46], and also AD patients in this study showed a high relative sequence read abundances of members of the genus *Staphylococcus* (Appendix A), which was correlated to the SCORAD (Figure 1). This high relative sequence read abundances in the microbiome is in contrast to microbiomes of non-AD skin patient, which reported lower sequence read abundances for *Staphylococcus* of 20 to 40% [47]. In addition, the other abundant genera *Enhydrobacter*, *Corynebacterium*, *Acinetobacter*, *Micrococcus* and *Streptococcus* commonly also occur as part of the human skin microbiome [47]. Therefore AD patients of this study showed a common skin microbiome with increased relative abundance of members of the genus *Staphylococcus*, which has been reported frequently before [20,21]. As the 16S rRNA gene sequence approach of the V3-V4 regions has a low phylogenetic resolution on species level for the skin microbiome [48], this study cannot differentiate shifts between *S. aureus* and other *Staphylococcus* spp. such as *S. epidermidis* or temporal/treatment specific shifts within the genus *Staphylococcus*.

In addition, the OTUs affiliated to the genus *Staphylococcus* showed no significant interactions (edges) in any bath treatment of the co-occurrence networks as these OTUs were always present and which do not reflect shifts in co-presence or mutual exclusions. Aside from members of the genus *Staphylococcus* relative sequence read abundance of AD patients after daily prebiotic and synbiotic bath treatment was shown to impact the composition of the microbiomes (Figure 2) as well as interaction patterns between members of the respective bacterial communities (Figure 3). Future research should also address the effect of probiotic treatment without prebiotics.

AD patients after daily synbiotic and prebiotic baths had a significantly decreased SCORAD and clearly visible improvement of their skin complexion and all QoL parameters (Table 1). Therefore, the application of the prebiotic components such as maltodextrin, inulin and apple pectin were beneficial for skin appearance, which has been shown for inulin also recently in another study [49]. However, the other prebiotic components were so far not studied in detail but may serve as nutrient for the beneficial skin microbiome. Moreover, skin dryness and pruritus was significantly better for AD patients after daily synbiotic baths. The bacterial microbiome showed an increase of the relative sequence reads of the genera *Lactobacillus* and *Bifidobacterium*, which were viable ingredient of the bath, only for AD patients treated with a synbiotic bath. The four strains from the genus *Lactobacillus* were described to modulate the host immune system through cytokine gene expression and to stimulate the phagocytosis by peripheral blood leucocytes [50]. In addition, both *Bifidobacterium* and all four *Lactobacillus* strains of the bath ingredient are potential bacteriocin producers and therefore able to limit the growth of opportunistic pathogens [51,52,53,54,55]. Moreover member of the genus *Bifidobacterium* can stimulate human immune function [56]. Furthermore, relative sequence read abundance of *Corynebacterium* decreased in the microbiome after synbiotic baths. Members of the genus *Corynebacterium* were associated with AD severity in the past [57]. Interestingly, OTUs affiliated to the order *Corynebacteriales* exhibited less interactions with other members of the bacterial microbiome with increasing treatment time (Appendix A) indicating a decreasing importance in the co-occurrence network. Moreover, Kwaszewska and colleagues showed a synbiotic relationship between *S. aureus* and members of the genus *Corynebacterium* due to a lack of proteinase activity [58]. Members of the genus *Brevibacterium* had also a decreasing topological role over time, and were reported as human pathogens capable to cause skin infections [59]. In turn, members of the genera *Acidibacter* and *Micrococcus* were described as module hubs from AD patients after synbiotic baths. While members of the genus *Acidibacter* were rarely found on human skin environments, *Micrococcus* is in top ten of the most frequent species of healthy human skin microbiomes [47] and is present only in low abundances in the skin microbiome of AD patients [60].

## 5. Conclusions

In conclusion, the treatment of AD patients after a daily synbiotic and to a lesser extend with a daily prebiotic bath decreased significantly the SCORAD and progressively improved QoL parameters. In addition, skin microbiome of AD patients after a synbiotic bath were colonized by strains from the probiotic bath. These strains have the generally recognized as safe (GRAS) status as probiotics and our results indicate that these strains are also harmless for skin applications. Taken together, this proof-of-concept study showed that a daily synbiotic bath, which included safe lactic acid bacteria strains, is a promising topical skin application to alleviate AD. Upcoming research should include more patients and address shifts in the host-microbe interaction of AD patients treated with synbiotic baths.

## Figures and Tables

**Figure 1 microorganisms-09-00527-f001:**
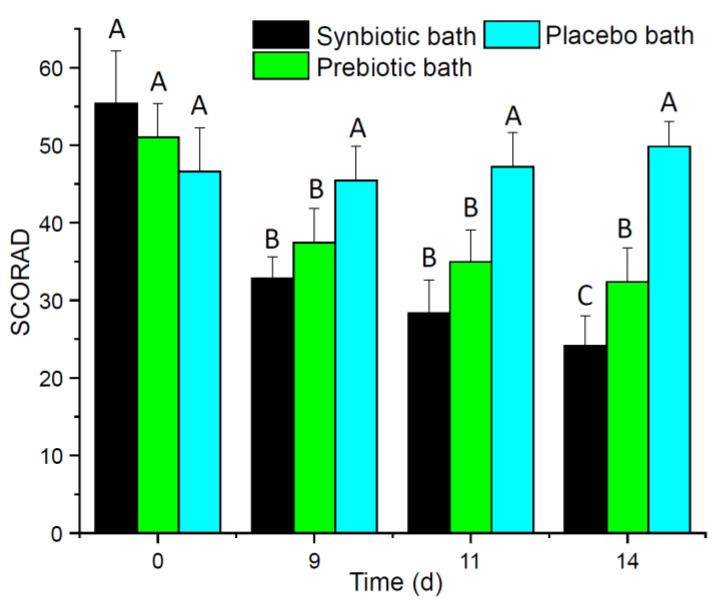
Severity scoring of atopic dermatitis (SCORAD) of atopic dermatitis (AD) patients after daily synbiotic (black bar, *n* = 7), prebiotic (green bar, *n* = 8) or placebo bath (cyan bar, *n* = 7) over time. Error bars indicate standard deviation. Different letters above bars within panels indicate significant differences (*p* < 0.05) according to two-way analysis of variance (ANOVA).

**Figure 2 microorganisms-09-00527-f002:**
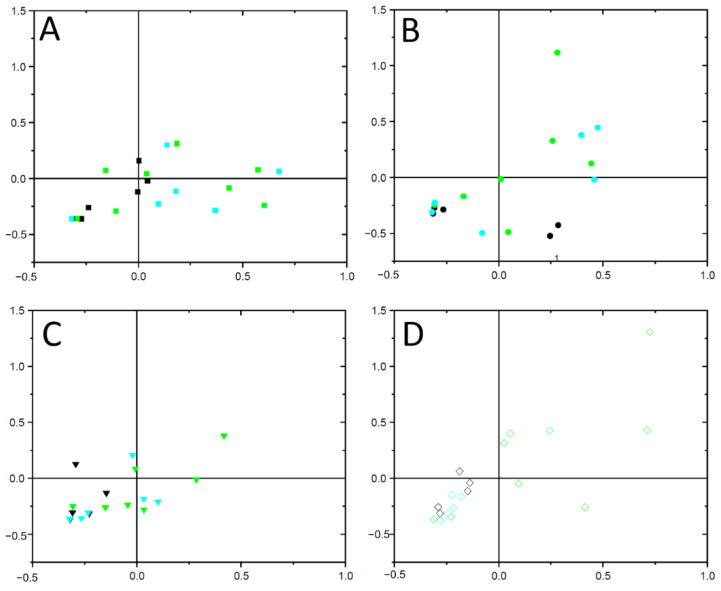
Correspondence analysis of the bacterial skin microbiome retrieved from AD patients after daily synbiotic (black), prebiotic (green) or placebo bath (cyan) before start (**A**, ■), and after 9 (**B**, ●), 11 (**C**,▼) and 14 days of bath treatment (**D**, ◊). The eigenvalues of the 1st and 2nd axes were λ_1_ = 0.53 and λ_2_ = 0.48.

**Figure 3 microorganisms-09-00527-f003:**
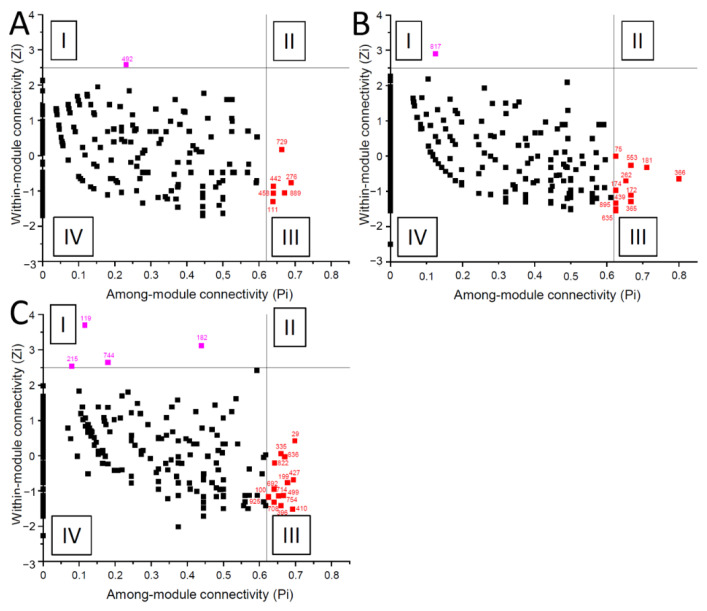
Network roles of each OTU of the bacterial skin microbiome retrieved from AD patients after daily synbiotic (**A**), prebiotic (**B**) or placebo bath treatment (**C**). Each OTU was categorized into module hubs (I), network hubs (II) connectors (III) or peripherals (IV) according to Olesen et al. (2007) [29]. OTU details of I-III can be found in Table 2 and OTU details of IV can be found in Appendix A.

**Table 1 microorganisms-09-00527-t001:** Two-way analysis of variance (ANOVA) of quality of life (QoL) parameters and severity scoring of atopic dermatitis (SCORAD) from AD patients after 14 days of daily synbiotic, prebiotic or placebo bath treatment. Significance of bath treatments, time or the interaction of bath treatments and time is highlighted in **bold** (*p* < 0.05). DF, degree of freedom.

Parameter	DF	F Value	*p* Value	DF	F Value	*p* Value
	***Overall assessment***			***Pain***		
**Time**	2	282.99	0.07	2	116.54	0.32
**Bath treatment**	**2**	**559.11**	**<0.01**	**2**	**522.90**	**<0.01**
**Bath treatment × Time**	4	0.77	0.55	4	0.27	0.90
	***Restriction***			***Lichenification***		
**Time**	2	15.76	0.22	2	183.79	0.17
**Bath treatment**	**2**	**688.58**	**< 0.01**	**2**	**147.13**	**<0.01**
**Bath treatment × Time**	4	0.25	0.91	4	0.66	0.62
	***Pruritus***			***Dryness***		
**Time**	**2**	**416.52**	**0.02**	**2**	**89.01**	**<0.01**
**Bath treatment**	**2**	**759.94**	**<0.01**	**2**	**921.14**	**<0.01**
**Bath treatment × Time**	4	0.92	0.46	4	136.97	0.26
		***Redness***		***SCORAD***		
**Time**	2	0.92	0.40	**2**	**889.67**	**<0.01**
**Bath treatment**	**2**	**986.83**	**<0.01**	**2**	**1060.98**	**<0.01**
**Bath treatment × Time**	4	0.68	0.61	**4**	**408.78**	**<0.01**

**Table 2 microorganisms-09-00527-t002:** Taxonomic classification and topological role of OTUs of the three co-occurrence networks of the bacterial skin microbiome retrieved from AD patients after 14 days of daily synbiotic, prebiotic or placebo bath treatment. OTUs of the network were assigned to connectors, module hubs or network hubs according to Olesen et al. (2007) [29] (see also Figure 3).

Treatment	OTU Number	Module	Taxon
**Synbiotic bath**	889	connector	*Acidibacter*
	442	connector	*Lachnoanaerobaculum*
	492	module hub	*Micrococcus*
	111	connector	*Brevibacterium*
	729	connector	*Conexibacter*
	276	connector	*Dermabacter*
	458	connector	*Leptotrichia*
	456	connector	*Pediococcus*
	696	connector	*Rickettsiales*
**Prebiotic bath**	365	connector	*Bergeyella*
	366	connector	*Chryseobacterium*
	75	connector	*Geobacillus*
	172	connector	*Alloiococcus*
	181	connector	*Brevundimonas*
	174	connector	*Carnobacterium*
	895	connector	*Dialister*
	439	connector	*Howardella*
	553	connector	*Marinomonas*
	817	module hub	*Obscuribacterales*
	262	connector	*Ohtaekwangia*
	635	connector	*Propionibacterium*
**Placebo bath**	499	connector	*Actinoplanes*
	119	module hub	*Aquabacterium*
	215	module hub	*Chthoniobacter*
	410	connector	*Janibacter*
	754	connector	*Jeotgalicoccus*
	29	connector	*Ilumatobacter*
	836	connector	*Acidobacteria*
	100	connector	*Bifidobacteriaceae*
	182	module hub	*Caulobacter*
	396	connector	*Hydrogenophilus*
	335	connector	*Peptoniphilus*
	925	connector	*Rhodanobacter*
	692	connector	*Rhodospirillales*
	708	connector	*Ruminococcaceae*
	822	connector	*Saccharimonadales*
	744	module hub	*Sphingopyxis*
	714	connector	*Subdoligranulum*

## Data Availability

The bacterial 16S rRNA gene sequences were deposited in the NCBI nucleotide sequence databases under accession PRJNA664092.

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
