# Peer review of "Improvement of Atopic Dermatitis by Synbiotic Baths"

_microorganisms, 2021, doi:10.3390/microorganisms9030527_

Round 1

Reviewer 1 Report

This paper is very interesting. I have only one suggestion: Figure 1 - more contrast colours can be used.

Author Response

This paper is very interesting. I have only one suggestion: Figure 1 - more contrast colours can be used.

Response: Thank you for this helpful comment! We have exchanged the black-whute figure with a color figure according to the color code of figure 2 in the revised version of the manuscript.

Reviewer 2 Report

The study may be interesting as proof of concept study, but it should be recognized that it has important drawbacks. First, the very  limited number of patients; second, the lack of a control group. Both of them call in question the conclusions. Methodological  limitations  need to be clearly stated by the authors

Major comments:

Study design

-How were the patients randomized? Could the authors give some details?

-a lyophilized powder was used as vehicle of pro and prebiotics. It is not clear how much powder (in grams) was added to the bath and which dilution rate was recommended to use to patients. The authors should clarify this important issue, in order to make the experiment reproducible

Results:

-Table 1 is very difficult to follow to the readers. It should be better to simplify data shown

-Discussion and conclusions: considering the limitation of the study, conclusion may be only “suggestive”, as this study should be considered only as a proof of concept study. Methodological  limitations  need to be clearly stated by the authors in the Discussion

Minor comments:

Introduction:

.Another relevant limitation to the use of TCS use is corticophobia, which may represent an obstacle to adequate applications of the correct TCS, worsening therapeutic compliance and limiting the possibility of a better AD control.( see and cite the paper by Chiricozzi A, Comberiati P, D'Auria E, Zuccotti G, Peroni DG. Topical corticosteroids for pediatric atopic dermatitis: Thoughtful tips for practice. Pharmacol Res. 2020 Aug;158:104878)

-Studies indicate a reduction of AD occurrence and severity after oral probiotic treatments in infants and adults by a systematic treatment [23,24]. References should be updated:  more recent systematic reviews show contrasting results on probiotics (oral) supplementation and atopic dermatitis  prevention (see and cite : the first metanalysis by Osborn 2007, followed by Lee J et al JACI 2008, more recent Cuello Garcia JACI 2015 and so on..).

-The study may be interesting as proof of concept study, but it should be recognized that it has important drawbacks. First, the very  limited number of patients; second, the lack of a control group. Both of them call in question the conclusions. Methodological  limitations  need to be clearly stated by the authors

Author Response

The study may be interesting as proof of concept study, but it should be recognized that it has important drawbacks. First, the very limited number of patients; second, the lack of a control group. Both of them call in question the conclusions. Methodological  limitations  need to be clearly stated by the authors

Response: Thank you for your interest in our study. We have included a control group, which was treated with autoclaved sand “placebo bath” instead probiotics or prebiotics and samples from this control group was analysed in the same way as the other treatments. We have had in the beginning of our experiment more patients but as the voluntary participation without any financial support for the patients, we have lost in each group 3 to 4 patients. For upcoming studies we will offer some money for compensation to come to our university and spend the time for our analyses.

Major comments:

Study design

-How were the patients randomized? Could the authors give some details?

Response: Thank you for this helpful comment. We have assessed the severity of each patient and thereafter patients with similar severity were randomly assigned to each group. We have not made any pre-selection based other criteria (age, sex,…). We have added this information in the material and methods section of the revised manuscript.

-a lyophilized powder was used as vehicle of pro and prebiotics. It is not clear how much powder (in grams) was added to the bath and which dilution rate was recommended to use to patients. The authors should clarify this important issue, in order to make the experiment reproducible

Response: Sorry for short our description. We have added additional information in the material and methods section of the revised manuscript how many grams contained which concentration, and how many Litres were used for legs and arms.  

Results:

-Table 1 is very difficult to follow to the readers. It should be better to simplify data shown

Response: Thank you for this useful hint. We deleted sum of squares and mean of squares, while only most important ANOVA-parameters (df, F and p Value) were retained in the revised manuscript. Furthermore, the values were rounded to two decimal places and the layout was changed to facilitate the readability of table 1.

-Discussion and conclusions: considering the limitation of the study, conclusion may be only “suggestive”, as this study should be considered only as a proof of concept study. Methodological  limitations  need to be clearly stated by the authors in the Discussion

Response: Thank you for your critical but constructive input. We have added in the conclusions of the revised manuscript version the information that this is a proof-of-concept study and that more patients should be included in upcoming research.

Minor comments:

Introduction:

.Another relevant limitation to the use of TCS use is corticophobia, which may represent an obstacle to adequate applications of the correct TCS, worsening therapeutic compliance and limiting the possibility of a better AD control.( see and cite the paper by Chiricozzi A, Comberiati P, D'Auria E, Zuccotti G, Peroni DG. Topical corticosteroids for pediatric atopic dermatitis: Thoughtful tips for practice. Pharmacol Res. 2020 Aug;158:104878)

Response: Thank you for this very helpful tip, which was directly included in the revised introduction.

-Studies indicate a reduction of AD occurrence and severity after oral probiotic treatments in infants and adults by a systematic treatment [23,24]. References should be updated:  more recent systematic reviews show contrasting results on probiotics (oral) supplementation and atopic dermatitis  prevention (see and cite : the first metanalysis by Osborn 2007, followed by Lee J et al JACI 2008, more recent Cuello Garcia JACI 2015 and so on..).

Response: Thank you again for this important information, which was also included in the revised introduction.

-The study may be interesting as proof of concept study, but it should be recognized that it has important drawbacks. First, the very  limited number of patients; second, the lack of a control group. Both of them call in question the conclusions. Methodological  limitations  need to be clearly stated by the authors

Response: As stated above we have addressed these concerns in the conclusions section of the revised manuscript.

Reviewer 3 Report

  1. Fig. 3A has one module hub, but the Table 2 has 2 module hubs in the part “Synbiotic bath”
  2. Supplementary Table S2 and Supplementary Table 3 were not attached as the Seperate files, therefore, the reviewer does not have the opportunity to get acquainted with them

3. Section 3.3 is poorly discussed by the authors. The composition of the microbiome in three groups, the role of modulators and connectors require a closer analysis

Author Response

  1. Fig. 3A has one module hub, but the Table 2 has 2 module hubs in the part “Synbiotic bath”

Response: Thank you for your careful reading. Yes, we have made an error in the table 2 which has been corrected in the revised manuscript version.

  1. Supplementary Table S2 and Supplementary Table 3 were not attached as the Seperate files, therefore, the reviewer does not have the opportunity to get acquainted with them

Response: We are sorry for this inconvenience. We have attached these both tables in the revised manuscript and hope it will be also available for the reviewers.

  1. Section 3.3 is poorly discussed by the authors. The composition of the microbiome in three groups, the role of modulators and connectors require a closer analysis

Response: The reviewed version of the manuscript already addressed the main biological findings of the networks in the last part of the discussion. As our discussion is written in a concise manner, additional extension with statistical explanation and not many new biological findings, will unease the reading flow. Therefore, we were interested not to add additional information.

Reviewer 4 Report

The manuscript entitled “Improvement of Atopic Dermatitis by Synbiotic Baths” by Noll et al. intends to evaluate by in in-vivo assays coupled to molecular biology techniques the impact of symbiotic and prebiotic baths on AD. The manuscript in very interesting and in the scopus of the journal. The paper is well written, and the materials & methods section well described. The results are if utmost importance for AD patients. I just have 2 minor concerns:

- why the authors did not tested only probiotics too?

- why the authors selected day 9, 11 and 14 for sampling?

- line 187: correct “sme“

Author Response

- why the authors did not tested only probiotics too?

Response: Thank you for this helpful comment. We have had only a very limited amount of voluntary patients and therefore we could only conduct three treatments (synbiotic, prebiotic and placebo treatment). We have added in the discussion section of the revised version a sentence that probiotics should be addressed in future approaches. Moreover, it is difficult to apply probiotics in adequate amounts without a “filler” or a carrier material. We have used prebiotics as a carrier for probiotics because prebiotics showed no adverse side effects on atopic skin. However, other carrier materials such as sugar, salt or flour has to be tested first, if they may also cause undesirable side effects.

- why the authors selected day 9, 11 and 14 for sampling?

Response: Again a very good question, which we have addressed in our discussion. In the revised version of the manuscript we have added the following information in the first paragrpah: This rebound phenomenon affected the early phase of each bath treatment as the SCORAD after three and seven days was not significantly different to the initial SCORAD. Therefore, we chose 9, 11 and 14 days of each bath treatment for in depth microbiome analyses.

- line 187: correct “sme“

Response: Thank you for your careful reading. We have corrected that typo in revised manuscript version.

Round 2

Reviewer 2 Report

No further comments

Some minor English typo errors 

Author Response

Some minor English typo errors.

Repsonse: We have carefully read the manuscript again, and we found few typo errors. These typo errors were corrected in the revised manuscript version.